# Radiolabeled NGR-Based Heterodimers for Angiogenesis Imaging: A Review of Preclinical Studies

**DOI:** 10.3390/cancers15184459

**Published:** 2023-09-07

**Authors:** György Trencsényi, Gábor Halmos, Zita Képes

**Affiliations:** 1Division of Nuclear Medicine and Translational Imaging, Department of Medical Imaging, Faculty of Medicine, University of Debrecen, Nagyerdei St. 98, H-4032 Debrecen, Hungary; trencsenyi.gyorgy@med.unideb.hu; 2Department of Biopharmacy, Faculty of Pharmacy, University of Debrecen, Nagyerdei St. 98, H-4032 Debrecen, Hungary

**Keywords:** Aminopeptidase N (APN/CD13), angiogenesis, asparagine–glycine–arginine (NGR), heterodimer, preclinical, radiolabeled

## Abstract

**Simple Summary:**

The increasing prevalence of malignant diseases and related metastases warrants the need for the establishment of specific diagnostic vectors that ensure timely as well as selective tumor identification. Since the application of multivalent ligands serves as an effective way to increase the binding ability of peptide-based molecular vectors, multi-target imaging is emerging as an intensively investigated area of cancer research. Given the colocalization of asparagine-glycine-arginine (NGR tripeptide) motif with angiogenesis-related Aminopeptidase N (APN/CD13), radiolabelled NGR-containing bispecific compounds may be valuable tools in the molecular diagnostics of APN/CD13 overexpressing tumors. Overall, the establishment of vascular-homing heterobivalent probes would not only broaden the horizon of cancer diagnostics but also contribute to the ultimate goal of the establishment of personalized tumor imaging.

**Abstract:**

Since angiogenesis/neoangiogenesis has a major role in tumor development, progression and metastatic spread, the establishment of angiogenesis-targeting imaging and therapeutic vectors is of utmost significance. Aminopeptidase N (APN/CD13) is a pivotal biomarker of angiogenic processes abundantly expressed on the cell surface of active vascular endothelial and various neoplastic cells, constituting a valuable target for cancer diagnostics and therapy. Since the asparagine–glycine–arginine (NGR) sequence has been shown to colocalize with APN/CD13, the research interest in NGR-peptide-mediated vascular targeting is steadily growing. Earlier preclinical experiments have already demonstrated the imaging and therapeutic feasibility of NGR-based probes labeled with different positron emission tomography (PET) and single-photon emission computed tomography (SPECT) radionuclides, including Gallium-68 (^68^Ga), Copper-64 (^64^Cu), Technetium-99m (^99m^Tc), Lutetium-177 (^177^Lu), Rhenium-188 (^188^Re) or Bismuth-213 (^213^Bi). To improve the tumor binding affinity and the retention time of single-receptor targeting peptides, NGR motifs containing heterodimers have been introduced to identify multi-receptor overexpressing malignancies. Preclinical studies with various tumor-bearing experimental animals provide useful tools for the investigation of the in vivo imaging behavior of NGR-based heterobivalent ligands. Herein, we review the reported preclinical achievements on NGR heterodimers that could be highly relevant for the development of further target-specific multivalent compounds in diagnostic and therapeutic settings.

## 1. Introduction

Considering the ever-increasing prevalence of malignancies and associated metastatic diseases, the continuous development of target-specific imaging probes is of crucial importance for the timely and accurate diagnostic assessment of neoplastic diseases.

Nowadays, due the outstanding sensitivity and specificity, nuclear medical techniques are emerging as prominent diagnostic tools in modern medical imaging. In clinical and preclinical nuclear medicine, various imaging modalities, such as positron emission tomography (PET) or single-photon emission computed tomography (SPECT), are used to visualize and quantify physiological processes at the molecular and cellular level using radiopharmaceuticals. Radiopharmaceuticals are radioactive substances—containing a radioisotope attached to a biologically active molecule—that are specifically designed to target certain tissues or molecules (for example, enzymes, receptors or other cell surface molecules) in the human body. Emitting externally detectable radiation, radiotracers allow for the tracking of specific biological processes or the evaluation of drug efficacy. Overall, the application of preclinical nuclear medical modalities provides a unique opportunity for exploring disease mechanisms in real time, assessing drug safety and effectiveness, optimizing treatment regimes and investigating novel diagnostic and therapeutic approaches [1]. Therefore, radiolabeled agents designed for the selective identification of tumor cell surface molecules represent a breakthrough in cancer imaging and have attracted the interest of several research groups.

At first, radiolabeled monoclonal antibodies were used for selective tumor-homing [2]. Their auspicious tumor binding potential along with notably high specificity made antibodies valuable candidates for cancer imaging and anti-tumor therapeutic purposes [3,4]. Furthermore, the tumor retention time of labeled antibodies is long enough to cover an extended imaging period. Despite initial successes with B-cell lymphomas [5,6], their widespread clinical applicability was mainly hampered by unsatisfactory tumor penetration [7]. Prolonged circulation time derived from the high molecular weight of the molecules as well as rising expenses constitute other considerable drawbacks of their routine application [8,9]. Furthermore, the extremely immunogenic properties of antibodies can easily trigger serious immune responses that pose another issue to be addressed [10].

More recently, aiming to overcome the inherent limitations of antibodies, radiolabeled peptides with lower molecular volume have become embraced for cancer imaging purposes. The development of peptide-based molecular vectors targeting cancer-related biomarkers stands out as one of the most exhaustively investigated areas of research [11,12,13]. The schematic representation of the structure of target-specific peptide-based radiopharmaceuticals is demonstrated in Figure 1. Numerous peptide-conjugated radioprobes appeared to be applicable both at preclinical and clinical levels [14,15]. Given the fairly rapid elimination, outstanding tissue penetration, insignificant immunogenicity and toxicity, as well as the easy and economically viable production, peptides have become the cornerstone of the synthesis of receptor-targeting diagnostic molecules [3,16]. Prompt clearance from the blood allows for target-specific accumulation without meaningful off-target uptake, which leads to the acquisition of highly contrasted images. The peptide pharmacokinetics, binding properties, stability and elimination kinetics could be seamlessly optimized by biochemical structural alterations [16]. In addition, a broad array of chelators is available for the attachment of the peptide to the radiometal without exerting any effect on the interplay between the ligand and its biological target.

Although peptides are superior to antibodies in several facets, the use of peptide-containing ligands is not without shortcomings either. Compared to antibodies, peptides have somehow lower binding affinity and tumor-targeting capability [3]. Another major constraint is their short retention time [3]. Therefore, different approaches have been introduced to improve the target selectivity and specificity of peptide-based imaging compounds. Out of these techniques, the application of multivalent ligands serves as an effective way to increase the binding ability of peptide-containing vectors [17]. Using peptide multimerization, the simultaneous binding of several targeting molecules to numerous receptor sites could be achieved [17,18]. Hence, applying multivalent probes, a higher chance of covering all subtypes of the expressed receptors could be granted than in the case of single-receptor-targeting diagnostic compounds. These polyvalent interactions ensure more enhanced overall adhering ability compared to the monomeric ones. Moreover, the likelihood of diminished targeting efficacy associated with tumor heterogeneity and “binding site barrier” is lessened using multi-receptor-targeting ligands [19,20,21]. A multi-target approach along with the favorable characteristics of peptides render radiolabeled peptides of global interest for radiopharmaceutical development and centralized distribution.

Peptide homodimers and multimers—created by the attachment of two or more molecules of the same type—are becoming preferred means to enhance receptor binding capacity. Comparing the tumor-targeting affinity of peptide homodimers and homomultimers with the monospecific counterparts, earlier findings of Ye et al., Sharma et al., Liu et al. and Zhang et al. confirmed the superiority of the polyvalent peptides over the former ones [22,23,24,25]. Exploiting the advantages of multivalency effect, previous researchers experienced improved α_v_β_3_ receptor binding using dimeric, tetrameric and polymeric arginine–glycine–aspartate (RGD) homomultimers [26,27,28,29].

Apart from peptide homomultimers, however, heterodimeric and heteromultimeric peptides are also precious tools for molecular cancer diagnostics. Both monovalent and bivalent heterodimers contain two different monomeric targeting subunits that can recognize distinct types of receptors in multi-receptor overexpressing tumors [30,31]. Multiple and simultaneous ligand–receptor interactions result in higher molecular affinity and specificity than monomeric connections [30]. Moreover, the increase in ligand concentration at the disease site and the advanced binding properties also contribute to the better target affinity of the heterodimers relative to that of the monomer match [32,33,34]. By virtue of more improved functional affinity, images with more suitable tumor-to-background ratio could be attained with the heterodimer-containing probe [35]. Other notable advantages of the bispecific compounds include higher targeting efficacy and avidity [36]. The higher sensitivity of the dual-targeting agents compared to the monospecific domains could be attributable to their more suitable in vivo pharmacokinetics, prolonged circulation time and bispecific binding ability [37,38,39]. Therefore, given the multivalent binding properties and related enhanced tumor-targeting capacity of radiolabeled peptide heterodimers/multimers, they seem to be highly valuable in molecular targeting and in oncological diagnostic settings. This is confirmed by a wide range of preclinical studies that have already published encouraging results on heterobivalent ligands labeled with different imaging entities [35,40].

## 2. Aminopeptidase N (APN/CD13) and Asparagine–Glycine–Arginine (NGR) Motif

Aminopeptidase N (APN/CD13) is considered as a central molecular biomarker of angiogenesis. This exopeptidase is abundantly expressed on the surface of healthy cells, such as the brush border of the epithelial cells of the proximal renal tubules and the small intestines, the epithelial cells of the prostate gland and the canaliculi of the bile ducts, as well as mast cells, fibroblasts and smooth muscle cells [41,42,43,44,45,46]. The majority of the myeloid cells and angiogenic endothelial cells also show high APN/CD13 positivity [42,47,48,49]. Furthermore, substantial APN/CD13 expression can be encountered on the highly active endothelial cells of the tumor-associated vasculature [50].

APN/CD13 is massively overexpressed in several cancers; examples include but are not limited to breast, prostate, ovarian, thyroid, pancreatic, colorectal, non-small-cell lung cancer and malignant pleural mesothelioma [51,52,53,54,55,56,57,58]. Although the degree of APN/CD13 expression of the different tumors is not yet fully uncovered, we suppose that the receptor pattern and density show a continuous change depending on the tumor stage, progression and aggressivity. The level of hypoxia and angiogenic growth factors, such as basic fibroblast growth factor (bFGF), tumor necrosis factor α (TNF α) and vascular endothelial growth factor (VEGF), may also determine the expression of APN/CD13 on tumor cells [59]. We hypothesize that, in more advanced phases of tumorigenesis when angiogenesis becomes vital for tumor progression and developed vasculature is required to ensure tumor cell nutrition, APN/CD13 is upregulated on the tumor cell surface. In later stages of tumor dedifferentiation, when necrosis and fibrotic transformation dominate, APN/CD13 expression is supposed to be downregulated. Based on the observations of Pasqualini et al. and Luan et al., however, the APN/CD13-mediated angiogenic processes occur regardless of its presence on the tumor cells; rather, these functions are regulated by the receptors presented on the tumor blood vessels [60,61]. In addition, the promoters that mediate APN transcription differ from cell to cell, and this could also be attributable to the differences between the receptor pattern of the tissues [62]. Since APN/CD13 plays a pivotal role in angiogenic processes, molecular probes addressing APN/CD13 could be strong applicants for the imaging of cancer-related angiogenesis [63].

Prior literature data state that asparagine–glycine–arginine (NGR) peptides colocalize with APN/CD13 [64,65]. The mechanism by which different substrates bind to APN/CD13 has been intensively investigated. Based on literature data, the interaction between the glutamate residue (Glu350) of the catalytic site of APN/CD13 and the N-terminal amino acid of the substrate (NGR) is responsible for APN/CD13 to exert its activity [66]. For the catalysis of ANP/CD13, the development of hydrogen bonds between the glutamate groups and the N-terminal amine group that fix and orientate the substrate (NGR) to APN/CD13 is a prerequisite [67]. Since the biochemical properties of the NGR sequence (for instance, cCNGRC, cCVLNGRMEC, cKNGRE, GNGRG and NGRAHA) could also regulate receptor binding, exhaustive research has been centered around the assessment of the association between the structure of NGR peptides and their function, and, based on the results, the scaffold in which NGR is implanted may influence its receptor binding capability, target specificity as well as stability [68]. For example, disulfide constraint has been reported to increase the binding affinity and the targeting potential of cyclic CNGRC and linear GNGRG to APN/CD13 [69]. Moreover, flanking residues trigger the transformation of NGR to isoaspartate–glycine–arginine (isoDGR), which leads to the switch in receptor selectivity from APN/CD13 to α_ν_β_3_ integrin [70]. The modification of the flanking residues appears to enhance target specificity and affinity [68]. In addition, Curnis and co-workers stated that the APN/CD13 binding of NGR-containing complexes is dependent upon the expression of the tumor-specific subtype of APN/CD13 [49]. Therefore, the accurate investigation of the receptor expression pattern of tumors for the design of target-specific diagnostic as well as therapeutic probes seems to be a prerequisite. Despite these observations, however, the structural requirements that are warranted for the receptor–ligand interaction still remain to be fully elucidated.

The efficacy of an NGR tripeptide containing radiolabeled diagnostic probes in cancer imaging has been strengthened by several research groups [71,72,73]. Applying PET or SPECT techniques, Li et al., Máté et al., Satpati et al., Vats et al. and Ma et al. explored the in vivo imaging behavior and the organ uptake pattern of various NGR derivatives labeled with Copper-64 (^64^Cu), Gallium-68 (^68^Ga), Technetium-99m (^99m^Tc), Lutetium-177 (^177^Lu) and Rhenium-188 (^188^Re) [71,72,73,74,75,76]. Out of these radioisotopes, ^68^Ga is used most commonly for the labeling of peptide-based compounds. Its favorable physical properties (T_1/2_: 67.71 min, Eβ^+^_average_: 830 KeV, maximum β^+^ energy: 1.92 MeV, Iβ^+^: 89%, Eγ: 1077 KeV, Iγ: 3.2%), decay scheme and easy elution from the germanium-68/gallium-68 (^68^Ge/^68^Ga)-generator system render ^68^Ga of increasing interest for the construction of PET radiopharmaceuticals [77,78,79,80]. Despite these highly desirable characteristics, radiolabeling with ^68^Ga is not without downsides either. Given its relatively short half-life, the shipment of the radiometal to remote centers and the labeling of molecules with prolonged pharmacokinetics definitely provide obstacles and challenges [81]. In addition, due to the short half-life, the application of ^68^Ga is restricted to cover imaging of short duration; moreover, ^68^Ga-based PET examinations could generally only be accomplished in facilities with on-site generators. Furthermore, high positron energy and related image noise is also an issue of concern of ^68^Ga usage [80]. Therefore, the search for labeling isotopes other than ^68^Ga may become the focus of increasing interest.

Although, as far as we are aware, the exact radiopharmaceutical uptake mechanism via APN/CD13 is not yet fully understood, the findings obtained from prior studies may provide some explanations. Investigating how the tumor-selective targeting of APN/CD13 could be achieved, Curnis et al. observed that the myeloid and epithelial cells, as well as the cancer-related angiogenic blood vessels, express varying subtypes of APN/CD13 [49]. This tissue/disease-specific variability in APN/CD13 may stem from the different protein conformations, the differences between the glycosylation of the receptors and the interplay with other molecules in the tumor niche [49,59,82]. Based on the existence of various APN/CD13 isoforms, we presume that the mode of action, the activity and thus the radiotracer uptake of APN/CD13 are largely dependent upon the receptor subtype [59].

Aside from radiolabeled NGR monomers, bispecific compounds are under investigation. Although NGR-based heterodimers are still in their early stage of application in cancer diagnostics, pioneering results have already confirmed their power in molecular imaging [37,83].

In this review, we summarize the most recent preclinical achievements with NGR containing multi-targeting heterodimers in nuclear medical settings. Figure 2 provides an overview of NGR-based heterodimers targeting angiogenic biomarkers in the tumor niche.

## 3. Overview of Preclinical Achievements with NGR-Based Bispecific Imaging Probes

### 3.1. NGR-RDG-Based Heterodimers for Angiogenesis Imaging

A number of NGR-tripeptide-based heterodimers have been developed for the molecular imaging of cancer-related angiogenic processes [37,83]. Most commonly, NGR is coupled with RGD (arginine–glycine–aspartate), which selectively clings to α_ν_β_3_, which is another marker of neo/angiogenesis (shown in Figure 2) [84].

To test the dual-receptor targeting ability of NGR-RGD heterodimers in gynecological xenografts, the research group of Gai and Long used ^68^Ga for the radiolabeling of the bispecific compounds [37,83]. The details of their studies are summarized in Table 1.

Gai and colleagues proposed NGR in conjunction with RGD for the PET imaging of breast cancers [37]. For the assessment of the in vivo imaging behavior of the [^68^Ga]Ga-labeled probes, mice bearing MCF-7, MDA-MB-231, MDA-MB-468 and MX-1 breast [37] and SKOV3 and ES-2 human ovarian tumors [83] were subjected to positron emission tomography/computed tomography (PET/CT) examinations and ex vivo organ distribution studies (as shown in Figure 3).

Similar to the ex vivo data, renal elimination was evidenced by prominent in vivo [^68^Ga]Ga-NGR-RGD accretion in the kidneys and in the urinary bladder, which could be in accordance with the hydrophilic properties of the molecules. In the rest of the abdominal and thoracic organs, including the lungs, heart, liver, spleen and the intestines, both teams registered moderate radioactivity. Based on the uptake values, except for the MX-1 tumors, [^68^Ga]Ga-NGR-RGD seemed to be a potent diagnostic probe for the identification of all investigated tumor types (in particular, SKOV3, ES-2 and MCF-7). Representative PET/CT images of both breast- and ovarian-tumor-bearing mice are presented in Figure 3. The extent of tumor radiotracer uptake was in line with the level of the receptorial expression determined by prior Western blot analysis and immunohistochemistry. Given the highest APN/CD13 and integrin α_ν_β_3_ expression levels observed in MCF-7 and MDA-MB-231 tumors, respectively, the most elevated [^68^Ga]Ga-NGR-RGD accumulation was observed in these tumor types, indicating the target selectivity of the probe (MCF-7: 1.03 ± 0.08%ID/g and MDA-MB-231: 1.04 ± 0.06%ID/g). MDA-MB-468 xenografts featured with lower APN/CD13 and α_ν_β_3_ integrin expression exhibited more discrete tracer accretion (MDA-MB-468: 0.76 ± 0.10%ID/g), while MX-1 tumors without the presence of either receptor showed negligible radioactivity (0.41 ± 0.08%ID/g). The quantitative assessment of [^68^Ga]Ga-NGR-RGD uptake in the breast cancer xenografts is displayed in Figure 3. In accordance with the observations of Gai et al., Long et al. [37,83] also experienced a positive correlation between the radiopharmaceutical accumulation and the degree of receptor upregulation, and this was confirmed by the prominent [^68^Ga]Ga-NGR-RGD values both in α_ν_β_3_ overexpressing SKOV-3 (0.68 ± 0.03%ID/g) and APN/CD13-positive ES-2 (0.70 ± 0.17%ID/g) tumors. The quantification of [^68^Ga]Ga-NGR-RGD accumulation of the ovarian tumors is demonstrated in Figure 3. The firm association between the receptor expression and the radiotracer uptake confirms the receptor affinity and tumor-homing capability of the heterodimer, and it also highlights the importance of the evaluation of tumor receptor pattern for the selection of the most suitable diagnostic tracer. In addition, diminished uptake values experienced upon in vivo and ex vivo receptor suppression studies further validated the selectivity of the bispecific compound.

Slow washout from the tumor tissue enables delayed imaging, which is of crucial significance regarding the detection of hidden lesions [37]. In addition, the experienced faint non-target activity along with the prominent tumor uptake result in the acquisition of highly contrasted PET images, which constitutes another clear advantage of heterodimers. PET scans with hardly detectable background noise make the accurate delineation of the neoplastic masses from the healthy organs possible. Moreover, the discrete background activity favors the detection of small metabolically active lesions and their clear distinction from the nearby normal tissues.

Applying pulmonary as well as abdominal metastatic models, Gai et al. and Long et al. [37,83] confirmed the metastases detection potential of [^68^Ga]Ga-NGR-RGD as well (presented in Table 1). In accordance with the immunohistochemical findings, the increased [^68^Ga]Ga-NGR-RGD uptake of the metastatic pulmonary lesions in the MCF-7 tumor models showed the applicability of the heterodimer in metastasis identification. Consistent with the ex vivo PET imaging, prominent [^68^Ga]Ga-NGR-RGD accumulation was found in the peritoneum implantation metastases of the ovarian tumors as well as the hepatic metastases of the ES-2 cancers, further indicating the excellence of the bispecific molecule in metastasis diagnostics.

A notable difference between the two detailed studies was that, while Gai et al. [37] compared the imaging performance of the heterodimer with that of the monomer probes ([^68^Ga]Ga-NGR and [^68^Ga]Ga-RGD), in the other study, the imaging properties of [^68^Ga]Ga-NGR-RGD were compared with those of the gold standard PET tracer, 2-deoxy-2[^18^F]fluoro-D-glucose ([^18^F]F-FDG) (as shown in Table 1). According to the MCF-7 tumor uptake values, Gai et al. proved the superior selectivity of the heterobivalent probe over the two monomeric agents [37]. In agreement with the immunohistochemical findings, the improved targeting capability of the heterodimer was further authenticated by the higher tracer concentration of the dual receptor expressing MCF-7 tumor compared to the other breast cancer xenografts. Based on the better contrast and tumor-to-non-target ratios obtained with [^68^Ga]Ga-NGR-RGD PET scanning, the research group proved the supremacy of the heterodimer over [^18^F]F-FDG as well [83]. Moderate [^18^F]F-FDG uptake in the abdominal metastases accompanied by considerable off-target activity indicated that [^18^F]F-FDG was inferior to the heterodimer in terms of the visualization of the metastatic lesions [83]. Furthermore, using muscular inflammation models, Long and co-workers [83] confirmed the feasibility of [^68^Ga]Ga-NGR-RGD in differential diagnostic settings. Significantly lower [^68^Ga]Ga-NGR-RGD activity of the inflamed muscle relative to its [^18^F]F-FDG uptake indicated the potential of the heterodimer in distinguishing between neoplastic and inflammatory tissues. This result highlights that [^68^Ga]Ga-NGR-RGD-based imaging may help to overcome the limitations of [^18^F]F-FDG scanning in terms of reducing the false positive results derived from the increased radioactivity of the inflamed tissues and organs.

The differences between the experienced tumor uptake values of the two studies could possibly stem from the differences between the applied tumor types with varying receptorial expression and intratumor heterogeneity. Furthermore, methodological disproportions may also partly explain the reason behind this. Gai et al. [37] acquired the PET/CT images 30 min post-injection of 6.5–9 MBq of the heterodimeric tracer, whereas, in the study of Long et al. [83] imaging with [^68^Ga]Ga-NGR-RGD (4–5.5 MBq) took place a day after the [^18^F]F-FDG PET/CT examinations. However, in both studies, the PET evaluation of the metastases was conducted an hour after tracer administration.

Overall, Gai et al. [37] and Long et al. [83] concluded that the performance of [^68^Ga]Ga-NGR-RGD surpasses that of the monospecific peers as well as the gold standard [^18^F]F-FDG in the molecular imaging of gynecological neoplasms with APN/CD13 and α_ν_β_3_ co-expression. Beyond metastasis detection, [^68^Ga]Ga-NGR-RGD could also be used to mitigate the false-positive findings on [^18^F]F-FDG scans derived from inflammation-related uptakes or normal physiological activities. Therefore, these findings capitalize on the relevance of this dual-receptor-targeting agent as a more specific alternative to the currently used monospecific counterparts.

Similar to Gai et al., the group of Sun and colleagues [85] also compared the diagnostic potential of NGR-RGD heterodimer with its monospecific peers [85]. Assessing the microPET/CT images of female Ncr nude mice xenografted with BxPC3 human pancreatic adenocarcinoma tumors, Sun et al. [85] further validated the superiority of the heterodimeric tracer over the monomer compounds ([^68^Ga]Ga-NGR, [^68^Ga]Ga-RGD). Table 1 provides an overview of their research. This was in line with previous research findings of Gai et al. on the same tracer using breast cancer xenografts [37]. The in vivo uptake trend of the three assessed radiotracers in the BxPC3 tumors was identical to that of the MCF-7 breast cancers, with the highest uptake values detected for the dimer and the lowest for the NGR monomer [37,85]. Identical to the above-discussed, Sun et al. [85] also strengthened the renal route of clearance of the ^68^Ga-labeled compounds. Like Long et al., Sun and colleagues [85] also accomplished the comparison of the imaging characteristics of the heterodimer with those of [^18^F]F-FDG [85]. Using KCH mice with pancreatic ductal adenocarcinoma (PDAC), [^64^Cu]Cu-RGD-NGR could successfully delineate the neoplastic site; nonetheless, it could not be delineated with [^18^F]F-FDG, and this finding correlated well with the observations of Long et al., who also reported that the radiolabeled NGR-RGD tracer displayed better in vivo performance in tumor detection than [^18^F]F-FGD [83,85]. Hence, the diagnostic ability of the heterodimers regardless of the imaging label surpassed that of the conventional glucose analogue. In both studies, higher tumor-to-muscle ratios were recorded for the radiolabeled NGR-RGD agent than for the [^18^F]F-FGD with the highest values registered in the case of the PDAC tumors. Comparing the uptake of [^64^Cu]Cu-RGD-NGR in the PDAC-bearing KCH mice and in the healthy pancreas of the tumor-naïve small animals, Sun and co-workers [85] noted that the normal tissue could not be depicted with the heterodimer, which further strengthened the target selectivity of the [^64^Cu]Cu-labeled derivative. In addition, the diagnostic feasibility of the [^64^Cu]Cu-labeled NGR-RGD heterodimer was also proved in ex vivo studies applying pancreatic intraepithelial neoplasia (PanIN) carrying 1-year-old KC^ER^ genetically engineered mice (*p48-Cre^ER^*^±^; *LSL-Kras^G12D^*^±^), which showed rapid PanIN uptake coupled with prompt elimination from the major organs and the best tumor-to-organ rations 4 h post-injection [85,86,87,88]. In a similar fashion, in vivo microPET examinations further validated the adequacy of [^64^Cu]Cu-RGD-NGR in the identification of the pathological pancreas of KC^ER^ mice bearing PanIN, while it failed to detect normal pancreatic tissue.

Overall, this study revealed consistent results with former observations [37,83] regarding the supreme imaging behavior of the heterodimer compared to both the monomer sisters and [^18^F]F-FDG. Given the capability of [^64^Cu]Cu-RGD-NGR to distinguish between pathological pancreatic lesions and normal tissue, Sun et al. [85] also emphasized the efficacy of NGR-RGD heterodimers in differential diagnostic settings.

### 3.2. Radiolabeled NGR-Hyaluronic Acid (HA) Containing Multimodal Imaging Probes

More recently, intensive research has been centered around the investigation of the role of NGR tripeptides in the enhancement of the tumor-targeting potential of several tumor specific molecules [89]. In a study of Li et al. [89], cancer-related CD44-targeting hyaluronic acid (HA) was complexed with NGR to test the effect of the addition of an NGR motif on the tumor uptake of HA (as presented in Table 2). For in vivo and ex vivo characterization, the probe was both fluorescently tagged with Alexa Fluor 647 and radiolabeled with beta emitter 177-Lutetium ([^177^Lu]Lu-DOTA/Alexa647-HA100-N). The synergistic effect of the NGR tripeptide for tumor-homing was proved by fluorescence imaging, which revealed meaningful accumulation of the DOTA/Alexa647-HA100-N compound with the NGR motif in NCI-H292- and A549-lung-tumor-carrying BALB/c mice, while, on the contrary, the NGR lacking agent did not accumulate in either of the tumors (DOTA/Alexa647-HA100). Previous studies of Ouasti et al. and Shi et al.—dealing with the attachment of integrin affine ligands (RGD/Arg-Gly-Asp or tetraiodothyroacetic acid) to HA—reported consistent results to those of [89] Li and colleagues [90,91]. In compliance with [89] Li et al., improved CD44/α_v_β_3_ targeting ability was observed by [91] Shi and colleagues for HA and tetraiodothyroacetic acid (tetrac) carrying solid–liquid nanoparticles (TeHA-SLNs) compared to the monospecific match (HA or tetrac) in CD44 and α_v_β_3_ overexpressing tumor cells and blood vessels of B16F10-melanoma-tumor-bearing C57BL/6 mice [91]. Furthermore, evaluating the accumulation of HA-RGD and HA in a phagocytic cell model of J774.2 murine macrophages, Ouasti et al. demonstrated the simultaneous binding of the bispecific probe to CD44 as well as to α_v_ integrins [90].

As mentioned by Li et al., owing to the VEGF-C-induced aggravated metastatic potential of the NCI-H292 tumors and related higher angiogenic activity, DOTA/Alexa647-HA100-N showed prolonged retention and more elevated concentration in the NCI-H292 tumors than in the A549 ones [89,92]. Since human squamous cell carcinoma cells, including NCl-H292 pulmonary cancer cells, do not exhibit APN/CD13 [93], increased uptake of the NCI-H292 tumors could be attributed to the profuse tumor angiogenesis and associated APN/CD13 positivity of vascular endothelial cells, such as human umbilical vein endothelial cells (HUVEC) and human aortic endothelial cells (HAEC) [94]. This could be further supported by the results of the in vitro cellular uptake examinations, indicating no tracer accumulation in the NCI-H292 cells. In line with the fluorescence imaging studies, the in vivo uptake profile of the radiolabeled compound ([^177^Lu]Lu-DOTA/Alexa647-HA100-N) also showed higher tracer accretion in the NCI-H292 tumors (%ID/g: 1.91 ± 0.97) compared to the A549 ones (%ID/g: 0.24 ± 0.17). Exploiting the therapeutic beta emission of ^177^Lu, this novel NGR-HA complex could be introduced in the workup of CD13- and CD44-positive targeted tumor radiotherapy. Therefore, due to the therapeutic beta rays (E_βmax_ = 497 keV) and the diagnostic gamma co-emission (E_γ_ = 208 keV) of the radiometal, [^177^Lu]Lu-DOTA/Alexa647-HA100-N seems to be a strong applicant in theranostic settings as well [95,96]. The most important details of the study of [89] Li et al. are displayed in Table 2.

Owing to the negligible toxicity, immunogenicity and the seamlessly modifiable structure of HA, the use of HA polymers, nanomaterials, bioconjugates or scaffolds coupled with a target-enhancing NGR motif may be widely accepted for drug delivery and regenerative processes [97,98,99,100]. The schematic overview of NGR-HA molecules is demonstrated in Figure 2. Although the current literature is lacking in experience about [^177^Lu]Lu-labeled HA derivatives, preliminary results are available on an HA-containing smart drug delivery system labeled with ^177^Lu [101]. Trujillo-Nolasco et al. [101] elaborated the synthesis of [^177^Lu]Lu-labeled drug-engulfing nanoparticles (PLGA) carrying HA and methotrexate (MTX) for the targeted radiotherapy of rheumatoid arthritis ([^177^Lu]Lu-DOTA-HA-PLGA(MTX)). Moreover, considering the CD44 selectivity of HA, NGR-conjugated HA complexes may be potent tools in targeting CD44-positive malignancies, including breast and colorectal cancers [102].

### 3.3. Radiolabeled NGR-Based Fusion Proteins

The interest in using vascular homing molecules for the delivery of cytotoxic or proapoptotic drugs to tumor neovasculature is steadily growing [103,104,105,106,107]. In a prior study of [108] Persigehl et al., an NGR-based fusion protein containing the soluble mutant form of thrombogenic human tissue factor (truncated TF, tTF) was constructed for tumor growth inhibition (shown in Table 3) [108].

Similar to [108] Persigehl and co-workers, other researchers also dealt with the investigation of NGR- or RGD-peptide-directed delivery of tTF proteins [109,110,111,112]. Although fusion proteins do not strictly belong to heterodimers in a narrower sense, because of the fact that tTF-NGR was radiolabeled (Iodine-123/^123^I) to track its tumor accumulation as well as distribution, we thought it noteworthy to be mentioned in the present review. Applying ^123^I NanoSPECT/CT-Plus acquisition—performed in the course of, or right after tracer injection—meaningful [^123^I]I-tTF-NGR uptake was depicted in the U87 tumors, especially in regions with highly activated angiogenesis. Therefore, even though the 7-minute-long half-life of tTF-NGR makes its SPECT-based biodistribution analysis challenging, [108] Persigehl et al. demonstrated the feasibility of SPECT in the verification of the selective tumor-homing potential of the fusion protein and in the quantitative assessment of tTF-NGR uptake in U87-glioblastoma-xenografted CD-1 nude mice. Further analyses of the [^123^I]I-tTF-NGR uptake pattern confirmed the hepatobiliary as well as renal route of elimination.

Different imaging modalities, including ultrasmall superparamagnetic particles of iron oxide (USPIO)-enhanced MRI, contrast-enhanced ultrasound (CEUS) and fluorescence reflectance imaging (FRI), were used to study the tumor response to tTF-NGR. Mitigated tumor perfusion and blood volume along with intratumor bleeding validated the therapeutic effect of tTF-NGR, which was achieved by thrombosis generation and consequent tumor necrosis. In correlation with the histological observations, the thrombogenic potential of tTF-NGR was authenticated by USPIO-enhanced MRI, which displayed more considerable relative blood volume reduction in the treatment-administered group (0.71 ± 0.22%) than in the control (1.67 ± 0.9%) as early as 4–8 h post-administration. Moreover, CEUS pointed out a significant decrease in relative tumor blood perfusion in the medicated cohort, supporting the anti-tumor efficacy of the probe.

Identical to [108] Persigehl and co-workers, Ma et al. also tested the applicability of an NGR-based fusion protein in diagnostic and radiotherapeutic settings [72]. Rhenium-188 (^188^Re)-labeled anti-angiogenic vascular endothelial growth inhibitor (VEGI) complexed with NGR was used for the imaging and the tumor growth inhibition of CD13-positive HT1080-xenotransplanted mice (as demonstrated in Table 3). Consistent with the results of [108] Persigehl et al., Ma et al. [72] further verified the feasibility of radiolabeled fusion proteins in simultaneous tumor diagnostics and therapy. SPECT data revealed the vascular homing selectivity of [^188^Re]Re-NGR-VEGI, which was in line with the post-imaging biodistribution results as well as the blocking studies. High-contrast SPECT images were acquired with notable tumor and minor non-target activity that are of utmost importance regarding lesion detection. While the HT1080 tumors could be visualized at all investigated time points (1, 4, 12 and 24 h post-tracer-injection), in another study of Ma and co-workers, 4T1 breast cancers were identifiable 4 h post-administration of the same tracer [72,113]. Ex vivo organ distribution studies also strengthened suitable tumor-to-background ratios, with the respective values being 4.98 ± 0.25, 0.61 ± 0.38 and 1.09 ± 0.31 for the tumor-to-muscle, tumor-to-kidneys and tumor-to-liver values [72]. In accordance with the observations of Persigehl et al. on [^123^I]I-tTF-NGR, the high renal (2.72 ± 0.48%ID/g) and hepatic (1.51 ± 0.46%ID/g) uptake of [^188^Re]Re-NGR-VEGI underpinned the primary role of the kidneys and the hepatobiliary system in radiopharmaceutical elimination. On the contrary, however, in the study with 4T1 xenografted female BALB/c mice, the urinary system represented the major way of clearance for [^188^Re]Re-NGR-VEGI [113]. The outstanding tumor inhibitory effect of [^188^Re]Re-NGR-VEGI proved the efficacy of the probe in targeted radiotherapy, which was in correlation with an earlier study of the same group using [^188^Re]Re-NGR-VEGI for the treatment of 4T1 breast tumors [72,113]. Consequently, in agreement with [108] Persigehl and colleagues, Ma and co-workers [72] also strengthened the suitability of NGR in the targeted transport of anti-tumor agents. Significant tumor growth inhibition was attained in the treated animals within a week after the administration of one regime of [^188^Re]Re-NGR-VEGI (18.5 MBq) in comparison with the treatment-naive cohort (*p* < 0.05). Further indicating the tumor cell inhibitory effect of the [^188^Re]Re-labeled vector, flow cytometry analyses showed that the [^188^Re]Re-NGR-VEGI-triggered HT1080 cell death was more pronounced compared to the control groups. The lack of untoward systemic toxicity favors the validation and integration of the probe for clinical application. Table 3 provides an overview of the studies of Ma et al. [18,72].

Based on these results, we may draw the conclusion that monitoring the distribution of anti-cancer drugs during application is of crucial importance to better assess treatment response and to modify treatment regime if indicated. Hence, we expect to see more radiolabeled NGR-based fusion proteins to be developed for molecular imaging in therapeutic scenarios.

Similar to Ma et al. [18,72] and Persigehl et al. [108], Vats and co-workers [76] also authenticated NGR fusion proteins as potent weapons in targeted cancer therapy [76]. Table 3 displays the most significant details of their study. Prior to radiolabeling, NGR tripeptide was fused with anti-tumor DNA-alkylating nitrogen mustard chlorambucil (CLB; [^99m^Tc]Tc-HYNIC-CLB-c(NGR).

Like the clearance of [^188^Re]Re-NGR-VEGI in 4T1 xenotransplants, the kidneys represented the main route of elimination in the case of [^99m^Tc]Tc-HYNIC-CLB-c(NGR) as well being consistent with the octanol partition coefficient [76,113]. On the contrary, previous studies with ^99m^Tc-tricarbonyl core-labeled CLB attached to chelators N_2_O or NNN atoms showed higher *LogP* values and corresponding increased gastrointestinal activity [114,115]. Thus, the addition of a targeting ligand to the drug could be exploited for the minimization of non-target image noise as well as cytotoxic side effects. In line with [^123^I]I-tTF-NGR, [^99m^Tc]Tc-HYNIC-CLB-c(NGR) also exhibited substantial tumor accumulation at early time points [76,108]. In addition, the tumor-to-muscle ratios of [^188^Re]Re-NGR-VEGI and [^99m^Tc]Tc-HYNIC-CLB-c(NGR) were largely comparable [72,76]. To further validate the in vivo target selectivity of the probe, Vats and colleagues [76] performed cell binding tests and blocking studies with B16F10 cells, the results of which also showed the receptor affinity of the radiolabeled derivative. Supporting the superior toxicity of the complex, the in vitro cytotoxicity studies revealed more augmented anti-tumor effects for the NGR-CLB conjugate on B16F10 melanoma cells in comparison with solo CLB or NGR. This projected the further application of NGR-CLB for in vivo therapeutic purposes. In a like manner, Gilad et al. also published a more enhanced tumor growth inhibitory effect using the RGD-CLB complex than in the case of the application of the RGD peptide without CLB or the chemotherapeutic agent alone [116].

These promising initial results on NGR-based anti-tumor complexes may propel the construction and the introduction of additional NGR peptide–drug conjugates for the achievement of target-selective drug transport and toxicity in neoplastic tissues.

Aside from human fibrosarcoma, glioblastoma and melanoma xenografts, the imaging properties of radiolabeled NGR-based fusion proteins were tested in preclinical models of prostate cancer as well. Li et al. [117] performed in vitro cellular uptake studies and in vivo microPET imaging to attest to the diagnostic efficacy of a novel [^68^Ga]Ga-labeled NKL probe made up of NGR motif and pro-necrotic D-K_6_L_9_ peptide in 22Rv1 human prostate carcinoma cell line and corresponding tumor-bearing mice (shown in Table 3) [117,118]. Tumor cell uptake values of 3.15% ± 0.04—measured at 2 h post-incubation—coupled with the suitable in vivo tumor accretion confirmed the affinity of [^68^Ga]Ga-DOTA-NKL for selective APN/CD13-positive tumor-homing. Similar to the previous study of Ma et al., high tumor contrast to contralateral background (9.97 ± 1.90) allowed for PET images with improved contrast [72].

According to the detailed results with NGR fusion proteins, considerable tumor growth inhibition was achieved by Asoh et al. as well using the complex of pro-apoptotic human Bax112-192 and green fluorescent protein (GFP) linked to NGR motif for the treatment of HeLa tumor xenotransplanted BALB/c-nu/nu mice [119]. Halving of the tumor volume as early as a week post-administration of NGR/GFP/Bax112-192 protein asserted the relevance of the compound in targeted tumor therapy. More pronounced NGR/GFP/Bax112-192-induced cell death in human umbilical vein endothelial cells (HUVECs) than in HeLa cells—demonstrated by the in vitro cellular experiments—further validated the selective anti-tumor potential of the fusion protein.

As part of investigating the mechanism behind the selective tumor-homing of NGR-based molecules, Curnis et al. analyzed the biodistribution of radiolabeled murine NGR-TNF conjugate in C57BL/6 mice [49]. This is presented in Table 3. Further, 6 or 24 h following intraperitoneal injection of 0.052 MBq/μg and 0.067 MBq/μg murine [^125^I]I-NGR-TNF and [^125^I]I-TNF, respectively, into C57BL/6 mice, the radiopharmaceutical uptake pattern of the tissues was determined. Studying the tissue-to-blood ratios in several organs, such as the intestines, liver, spleen, bones (femur, sternum), the kidneys, the thoracic organs, muscle and the skin of the two derivatives, Curnis et al. [49] concluded that the radiolabeled NGR-TNF compound showed no accretion in normal tissues with APN/CD13 positivity. No significant difference was found between the organ-to-blood ratios of [^125^I]I-NGR-TNF and [^125^I]I-TNF, supporting the lack of binding of the fusion protein to normal tissues. Taking into account that the tumor-specific receptor subtype highly determines the APN/CD13 binding potential of NGR–drug complexes [49], the assessment of the tumor receptor profile prior to the development of NGR-based diagnostic or therapeutic compounds is required.

**Table 3 cancers-15-04459-t003:** Radiolabeled NGR fusion proteins.

Investigated Object	Aims	Radiopharmaceutical	In Vitro and/or In Vivo Methods	Reference
U87- or HT1080-xenotransplant-bearing athymic female nude mice (CD-1 nu/nu)	tTF-NGR treatment monitoring, performance evaluation of multi-modal vascular and molecular diagnostics	iodine-123-labeled tTF-NGR ([^123^I]I-tTF-NGR)	SPECT/CT imaging (NanoSPECT/CT-Plus), CEUS, USPIO-enhanced MRI, FRI with AngioSense^680^, tumor treatment studies with tTF-NGR, NGR-blocking experiments with blocking peptide GNGRAHA and tTF-NGR histology (H&E staining)	[108]
HT1080 cell line	radioimaging, radiotherapy	Rhenium-188-labeled NGR-VEGI ([^188^Re]Re-NGR-VEGI)	in vitro: stability, flow-cytometry-based apoptosis assay with vector control, NGR peptide, VEGI protein, purified NGR-VEGI protein and [^188^Re]Re-NGR-VEGI	[72]
HT1080-tumor-bearing female BALB/c nude mice	radioimaging, radiotherapy	Rhenium-188-labeled NGR-VEGI ([^188^Re]Re-NGR-VEGI)	in vivo: static SPECT acquisition, blocking studies with cold NGR-VEGI, ex vivo biodistribution studies, radiotherapy experiments (H&E staining, tumor size, body weight measurement)	[72]
in vitro	in vitro assessment, quality control, synthesis	Rhenium-188-labeled NGR-VEGI ([^188^Re]Re-NGR-VEGI)	in vitro stability	[18]
in vivo female BALB/c mice bearing 4T1 cancer	radioimaging, radiotherapy	Rhenium-188-labeled NGR-VEGI ([^188^Re]Re-NGR-VEGI)	radiotherapy experiments with [^188^Re]Re-NGR-VEGI, NGR-VEGI, NGR and saline control (tumor growth registration, H&E and (TUNEL) staining.	[18]
in vitro human B16F10 cell line	in vitro evaluation	[^99m^Tc]Tc-HYNIC-labeled CLB-c(NGR) ([^99m^Tc]Tc -HYNIC-CLB-c(NGR))	in vitro stability, cell-binding experiments, determination of specific uptake with peptide [c(KCNGRC)], MTT-assay-based in vitro cytotoxicity studies with the PDC, cNGR peptide and CLB	[76]
in vivo C57BL/6 mice xenografted with murine B16F10 cells	evaluation of APN/CD13 targeting ability	[^99m^Tc]Tc-HYNIC-labeled CLB-c(NGR) ([^99m^Tc]Tc -HYNIC-CLB-c(NGR))	in vivo biodistribution studies, blocking studies applying non-radiolabeled c(NGR)	[76]
in vitro 22Rv1 cells	in vitro characterization	[^68^Ga]Ga-DOTA-NKL	uptake rate	[117]
in vivo 22Rv1-tumor-bearing mice	assessment of tumor-targeting properties and APN/CD13 diagnostic potential	[^68^Ga]Ga-DOTA-NKL	in vivo microPET/CT imaging	[117]
C57BL/6 mice	investigation of tumor-targeting properties of NGR–drug complexes	iodine-125-labeled murine NGR-TNF and iodine-125-labeled murine TNF ([^125^I]I-NGR-TNF and [^125^I]I-TNF)	in vivo uptake pattern determination	[49]

APN/CD13: Aminopeptidase N; B16F10: murine melanoma cell line; CEUS: contrast-enhanced ultrasound; CLB: DNA-alkylating nitrogen mustard, chlorambucil; CT: computed tomography; DOTA: 1,4,7,10-tetraazacyclododecane-1,4,7,10-tetraacetic acid; FRI: fluorescence reflectance imaging; ^68^Ga: Gallium-68; H&E: hematoxylin and eosin; HT1080: human fibrosarcoma cell line; ^123^I: iodine-123; ^125^I: iodine-125; MRI: magnetic resonance imaging; NGR: asparagine–glycine–arginine; NKL: fused NGR and D-K6L9 peptides; PDC: peptide–drug conjugate; PET/CT: positron emission tomography/computed tomography; ^188^Re: Rhenium-188; SPECT: single-photon emission computed tomography; ^99m^Tc: Technetium-99m; TNF: tumor necrosis factor-alpha; tTF: truncated tissue factor, TUNEL: terminal-deoxynucleotidyl-transferase-mediated dUTP nick-end-labeling/TdT (terminal deoxynucleotidyl transferase)-mediated dUTP nick-end labeling; U87: human glioblastoma cell line; USPIO-enhanced MRI: ultrasmall superparamagnetic iron-oxide-enhanced magnetic resonance imaging; VEGI: vascular endothelial growth inhibitor; 4T1: mouse breast cancer cell line (triple-negative breast cancer (TNBC) cell line lacking in estrogen receptor (ER), progesterone receptor (PR) and human epidermal growth factor receptor 2 (HER2) expression); 22Rv1: human prostate cancer.

### 3.4. NGR Heterodimers beyond Nuclear Medical Applications

The use of NGR-based imaging probes is not strictly limited to the field of nuclear medicine. Initiated by prior results [120,121], Oostendorp et al. were the first to propose the construction of a cNGR containing an imaging entity for MRI-guided angiogenesis detection [122]. cNGR-labeled paramagnetic quantum dots (NGR-pQD) were used to quantitatively assess tumor angiogenesis with a molecular magnetic resonance imaging (MRI) technique in experimental tumor models [122]. As mentioned by [122] Oostendorp et al., cNGR coupled with pQD appeared to be more specific in the identification of tumor regions with the most enhanced angiogenic activity relative to pQDs alone. Fifteen-week-old male Swiss nu/nu mice bearing human colorectal adenocarcinoma (LS174T) were intravenously administered with either the CD13 targeting the NGR-pQD ligand or the pQDs without the NGR motif to test the feasibility of the complex in the assessment of tumor-related angiogenesis applying quantitative molecular MRI. Supported by elevated image contrast and longitudinal relaxation rate *R*_1_ (1/*T*_1_) with accompanying diminished proton visibility, the estimation of angiogenic activity as well as the visualization of tumor sites with meaningful neoangiogenesis is possible using NGR-pQD particles. The angiogenesis imaging specificity of the novel probe was further evidenced by the three times more elevated MRI contrast in highly angiogenic neoplastic regions as well as its minimal background concentration. Consistent with the MRI results, ex vivo two-photon laser scanning microscopy (TPLSM) also proved the selective binding of the NGR compound to the APN/CD13 receptors. Since MRI and TPLSM are not satisfactory for the thorough biodistribution assessment of NGR-pQD, novel approaches such as the PET technique could allow for a more accurate uptake analysis. Therefore, there is an increasing need for the development of NGR-based MRI agents carrying a positron emitter radiometal. Moreover, considering these promising results, it is therefore worthwhile to continue identifying and characterizing NGR-based molecules that can be utilized for the MRI-based detection of tumor-associated angiogenesis, with special emphasis on dual-targeting specific ligands.

## 4. Conclusions

The introduction of multi-receptor-targeting NGR-based radiopharmaceuticals heralds a new era in the diagnostics of cancer-related angiogenesis. Radiolabeled heterobivalent NGR-RGD tracers seem to be strong applicants for the improvement of tumor imaging in the foreseeable future, not only by enabling more precise lesion detection because of their multiple receptor binding capability but also by ensuring better contrast and improved signal-to-noise ratios that elucidate more about the association between the neoplastic lesions and the surrounding healthy tissues as well as metastases, making precise TNM tumor staging possible [83]. Given the multiple receptor adhering ability, the use of NGR-RGD probes may overcome the limitations of single-targeting agents regarding the delineation of heterogenous tumor masses or lesions with changing receptorial patterns. In addition, the identification of necrotic or fibrotic tumors with irregular receptor density and receptor distribution—in which monovalent probes generally fail—is also feasible upon multi-targeting imaging. ^68^Ga seems to be the optimal isotope for the radiolabeling of NGR-based heterodimers so far; however, given the suitable tumor uptake of ^64^Cu-labeled bispecific probes, ^64^Cu could be a viable alternative for synthesis processes. Based on the outstanding tumor-homing capability and the specificity of the detailed imaging entities, cKNGRE [37,83] is ideal for the construction of heterobivalent probes. Although no prior studies dealt with the examination of the effect of the different NGR peptides on the in vivo imaging behavior of an NGR motif containing multivalent agents, we suppose that the chemical structure of the NGR sequence—cyclic, linear, etc.—might alter the affinity, target specificity and the kinetics of the diagnostic molecules.

Beyond the NGR-RGD heterodimers developed so far, the discussed promising initial results fuel enthusiasm towards the proposal of other angiogenesis-targeting bispecific derivatives. However, the lessons learned from the couple of years of experience with radiolabeled NGR/RGD vectors have drawn our attention to the fact that several factors must be addressed to ensure the development of molecular entities with suitable pharmacokinetics as well as imaging properties [3,35]. First, to select the most appropriate targeting ligands, the exploration of the tumor receptor profile and receptor density is of foremost importance. Second, besides the physicochemical properties of the heterodimers, including size, charge, polarity, aromaticity, solubility or surface area, the type of the radiometal used for labeling is also a major factor of the in vivo imaging behavior [3,35]. In addition, emphasis must be placed upon the choice of the chelator or other linkers that do not impact either the biological behavior of the ligand and the receptor or the interaction between them. Considering these parameters is mandatory to achieve optimized pharmacokinetics, including reduced off-target noise, enhancement of tumor uptake and related improved tumor-to-background contrast. Overall, the development of further vascular-homing bispecific vectors would not only broaden the horizon of cancer research but also contribute to the ultimate goal of the establishment of personalized tumor imaging. Prior to their clinical transportation, however, comprehensive preclinical studies and clinical trials are warranted to validate their safety and imaging power.

Furthermore, given the intrinsic property of angiogenesis-targeting heterodimers to distinguish between activated and quiescent endothelial cells, the use of NGR heterodimers radiolabeled with alpha or beta emitter therapeutic isotopes is emerging as a potentially new approach to targeted cancer therapy. Providing a representation of a tumor receptorial pattern, the diagnostic match of the therapeutic heterodimer compound allows for the selection of those patients who would be privileged from antiangiogenic cancer treatment.

Finally, it is firmly believed that predictive biosignals derived from imaging with radiolabeled NGR-based heterodimers not only have outstanding potential for molecular cancer diagnostics but, more importantly, may lay the groundwork for the development of therapeutic counterparts for theranostic applications.

## Figures and Tables

**Figure 1 cancers-15-04459-f001:**
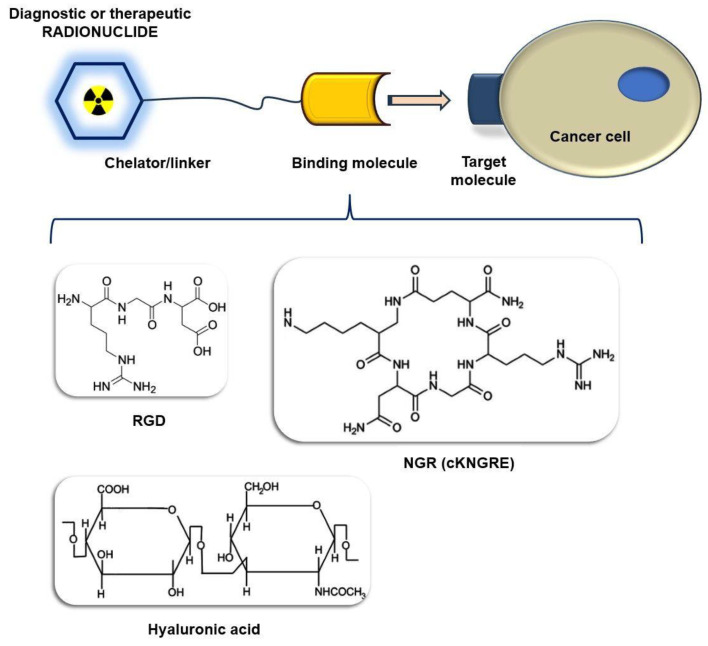
Schematic representation of the structure of target-specific peptide-based radiopharmaceuticals. This figure demonstrates the linkage of the targeting moiety and the radioisotope to form a radiopharmaceutical that specifically binds to a cancer-related target molecule. A radiopharmaceutical consists of a target-specific, biologically active binding molecule (e.g., NGR tripeptide, RGD) attached via a chelator/linker to a labeling radioisotope. The radioactive substances—radiotracers—selectively target biomarkers expressed on the surface of the tumor cells (e.g., APN/CD13, α_ν_β_3_ integrin). In diagnostic settings, a radionuclide with β^+^ or γ emission is bound to the peptide-chelator core, which allows for the in vivo assessment of biological processes at the molecular level using PET or SPCET techniques. On the other hand, radiolabeling with γ- or β-emitting therapeutic isotopes makes selective radiotherapy and tumor killing possible. The use of target-specific peptides as carriers of cytotoxic or proapoptotic drugs (e.g., tTF, VEGI, CLB) and tumor-affine molecules such as HA seems to be successful in the enhancement of their tumor-homing potential and cytotoxic effect.

**Figure 2 cancers-15-04459-f002:**
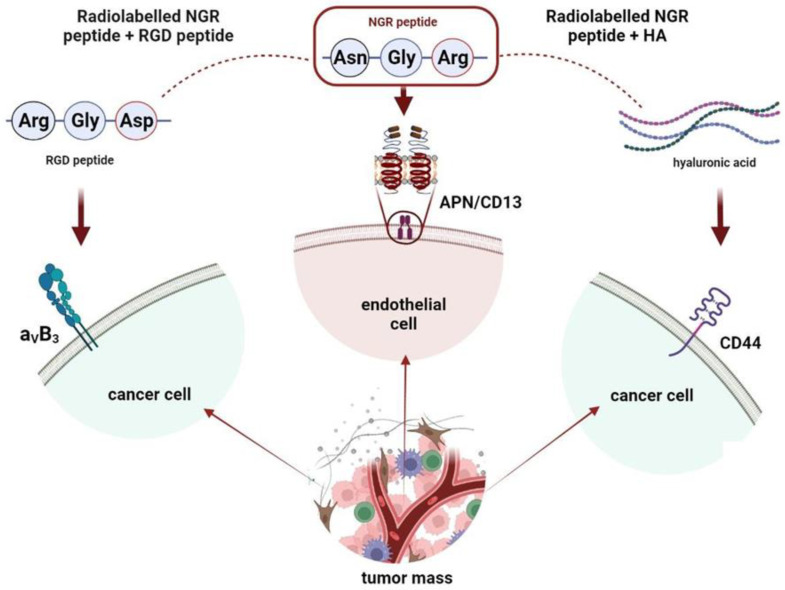
Schematic overview of NGR-based heterodimers targeting angiogenesis in the tumor microenvironment.

**Figure 3 cancers-15-04459-f003:**
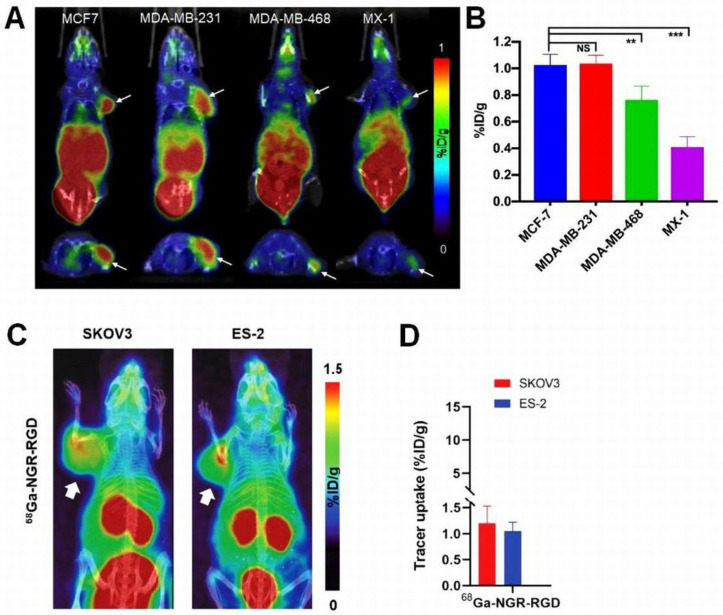
PET/CT imaging and quantitative assessment of [^68^Ga]Ga-NGR-RGD uptake in preclinical mouse models of breast and ovarian tumors. (**A**) Representative PET/CT images of MCF-7, MDA-MB-231, MDA-MB-468, and MX-1 breast cancer-bearing female BALB/c nude mice 30 min after the intravenous injection of 6.5–9 MBq of [^68^Ga]Ga-NGR-RGD. The white arrows point to the tumors. (**B**) Quantitative assessment of [^68^Ga]Ga-NGR-RGD uptake in MCF-7, MDA-MB-231, MDA-MB-468, and MX-1 tumors. The uptake values are obtained as mean %ID/g ± SD. (**C**) Representative PET/CT images of SKOV-3 and ES-2 ovarian tumor-bearing female BALB/c nude mice 1 h after the intravenous injection of 4–5.5 MBq of [^68^Ga]Ga-NGR-RGD. The white arrows demonstrate the localization of the subcutaneous tumors. (**D**) Quantitative evaluation of [^68^Ga]Ga-NGR-RGD accumulation in SKOV-3 and ES-2 xenografts. The results are expressed as %ID/g ± SD. ** *p* < 0.01, *** *p* < 0.001, NS: non significant. (**A**,**B**) With permission from [37]. (**C**,**D**) With permission from [83].

**Table 1 cancers-15-04459-t001:** Overview of preclinical studies with radiolabeled NGR-based heterodimers.

Investigated Object	Investigated Phenomenon/Initiatives	Radiopharmaceutical	In Vitro and In Vivo Methods	Reference
in vitro: MCF7, MDA-MB-231, MDA-MB-468 and MX-1 cell lines	synthesis and radiolabeling of NGR-based heterodimeric tracer ([^68^Ga]Ga-NGR-RGD), in vitro biological properties	[^68^Ga]Ga-NGR-RGD dimer, [^68^Ga]Ga -RGD monomer, [^68^Ga]Ga -NGR monomer	in vitro: cellular uptake studies, blocking studies (pretreatment with cold RGD, NGR, RGD+NGR or NGR-RGD peptides) other: Western blot analyses	[37]
in vivo: female BALB/c nude mice bearing MCF-7, MDA-MB-231, MDA-MB-468 and MX-1 tumors, pulmonary metastases mouse models	synthesis and radiolabeling of NGR-based heterodimeric tracer ([^68^Ga]Ga-NGR-RGD), in vivo diagnostic potential, APN/CD13 and integrin α_ν_β_3_ targeting capability, comparison with monospecific NGR and RGD tracers	[^68^Ga]Ga-NGR-RGD dimer, [^68^Ga]Ga -RGD monomer, [^68^Ga]Ga -NGR monomer	in vivo: microPET/CT acquisition, in vivo blocking studies with non-radiolabeled RGD, NGR, RGD+NGR and NGR-RGD peptides, post imaging biodistribution studies other: immunohistochemical analyses	[37]
in vitro: SKOV3, ES-2 and OVCAR4 cell lines	in vitro characterization	[^68^Ga]Ga-NGR-RGD dimer	in vitro cell uptake and blocking studies (pretreatment with non-radioactive NGR-RGD or NGR+RGD) other: Western blot analyses	[83]
in vivo: SKOV3 and ES-2 tumor bearing female BALB/C nude mice, in vivo abdominal metastatic models of ovarian cancer, muscular inflammation models	in vivo diagnostic performance, APN/CD13 and integrin α_ν_β_3_ targeting ability, efficacy in differential diagnostic settings, comparison with [^18^F]F-FDG	[^68^Ga]Ga-NGR-RGD dimer, [^18^F]F-FDG	in vivo PET/CT imaging, ex vivo organ distribution studies other: immunohistochemical analyses	[83]
BxPC3 tumor-carrying female Ncr nude mice, autochthonous mouse models bearing late stage PanIN lesions (KC^ER^ mice) or PDAC (KPC mice), KCH genetically engineered mouse model of PDAC	in vivo imaging behavior of the dimeric RGD-NGR compound, comparison with the monospecific peers and [^18^F]F-FDG	[^68^Ga]Ga-RGD-NGR dimer, [^64^Cu]Cu-RGD-NGR dimer, [^68^Ga]Ga-RGD monomer, [^68^Ga]Ga-NGR monomer, [^18^F]F-FDG	^68^Ga-based in vivo microPET/CT imaging (BxPC3 tumorous mice), ^64^Cu-based microPET/CT imaging (C57BL/6, KC, KPC, KCH mice), [^18^F]F-FDG PET/CT acquisition (KCH mice) ex vivo biodistribution studies with [^64^Cu]Cu-RGD-NGR (C57BL/6, KC, KPC, KCH mice), ex vivo blocking experiments with cold RGD and NGR peptides (KC mice) other: H&E staining, Immunofluorescence staining	[85]

APN/CD13: Aminopeptidase N; BxPC3: human pancreatic adenocarcinoma cell line; ES2: human ovarian cancer cell line; [^18^F]F: Fluorine-18; [^18^F]F-FDG: 18-Fluorine-2-fluoro-2-deoxy-d-glucose; ^68^Ga: Gallium-68; H&E: hematoxylin and eosin; MCF-7: human breast cancer cell line (estrogen (ER) and progesterone receptor (PR) positive); MDA-MB-231: human breast cancer cell line (triple-negative breast cancer (TNBC) cell line lacking in estrogen receptor (ER), progesterone receptor (PR) and human epidermal growth factor receptor 2 (HER2) expression); MDA-MB-468: human breast cancer cell line (triple-negative breast cancer (TNBC) cell line lacking in estrogen receptor (ER), progesterone receptor (PR) and human epidermal growth factor receptor 2 (HER2) expression); MX-1: human breast cancer cell line (estrogen (ER) negative); NGR: asparagine–glycine–arginine; PanIN: pancreatic intraepithelial neoplasia; PDAC: pancreatic ductal adenocarcinoma; PET/CT: positron emission tomography/computed tomography; RGD: arginine–glycine–aspartate; SKOV-3: human ovarian adenocarcinoma cell line; OVCAR4: human ovarian cancer cell line.

**Table 2 cancers-15-04459-t002:** Summary of radiolabeled NGR-HA containing multimodal imaging probe.

Investigated Object	Initiatives	Molecular Probes	In Vitro and In Vivo Methods	Reference
A549, HT-1080, MDA-MB-231, MCF-10A, HCT-116, NCI-H292 cell lines	development of multi-functionalized probe, in vitro characterization, in vitro cytotoxicity	DOTA/Alexa647-HA7, DOTA/Alexa647-HA100, DOTA/Alexa647-HA100-N, DOTA/Alexa647-HA7-N, HA100, HA7, [^177^Lu]Lu-DOTA/Alexa647-HA100-N	determination of cell viability by MTS assay, cellular uptake studies applying microscopic observation, in vitro stability, radiochemical purity measured by iTLC	[89]
NCI-H292- and A549-lung-tumor-carrying BALB/c mice	development of multi-functionalized probe, in vivo and ex vivo characterization, evaluation of CD44 and APN/CD13 targeting capability, therapeutic efficacy	177-Lutetium-labeled and Alexa Fluor 647-tagged NGR-HA complex ([^177^Lu]Lu-DOTA/Alexa647-HA100-N) DOTA/Alexa647-HA7-N, DOTA/Alexa647-HA100-N, DOTA/Alexa647-HA100	ex vivo/in vivo fluorescence imaging (with DOTA/Alexa647-HA7-N, DOTA/Alexa647-HA100-N and DOTA/Alexa647-HA100 for control) in vivo biodistribution ([^177^Lu]Lu-DOTA/Alexa647-HA100-N)	[89]

Alexa Fluor 647: fluorescent probe; APN/CD13: Aminopeptidase N; A549: human pulmonary adenocarcinoma cell line; CD44: cluster of differentiation 44; DOTA: 1,4,7,10-tetraazacyclododecane-1,4,7,10-tetraacetic acid; HA (HA100, HA7): hyaluronic acid; HCT-116: human colorectal carcinoma cell line; HT1080: human fibrosarcoma cell line; iTLC: instant thin-layer chromatography; ^177^Lu: 177-Lutetium; MCF-10A: human epitheloid breast cancer cell line (estrogen receptor (ER) α negative, estrogen receptor (ER) β negative); MDA-MB-231: human breast adenocarcinoma cell line (triple-negative breast cancer (TNBC) cell line lacking in estrogen receptor (ER), progesterone receptor (PR) and human epidermal growth factor receptor 2 (HER2) expression); MTS: 4,5-dimethylthiazol-2-yl)−5-(3-carboxymethoxyphenyl)−2-(4-sulfophenyl)−2-H-tetrazolium; NCI-H292: human mucoepidermoid pulmonary cancer cell line; NGR: asparagine–glycine–arginine.

## Data Availability

The datasets used and/or analyzed during the current study are available from the corresponding author upon reasonable request.

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
