# Peer review of "Radiolabeled NGR-Based Heterodimers for Angiogenesis Imaging: A Review of Preclinical Studies"

_cancers, 2023, doi:10.3390/cancers15184459_

Round 1

Reviewer 1 Report

In this review article entitled “Radiolabeled NGR-based Heterodimers for Angiogenesis Imaging: A Review of Preclinical Studies”, the authors wrote a comprehensive review on the Aminopeptidase N (APN/CD13) targeted molecular imaging using radiolabelled NGR-contained peptide ligands, with particular interest on heterodimeric radiotracers. In addition, the authors also discussed the opportunities on using NGR-based heterodimers for radiotheranostics. The manuscript was well organized, and the literature citation was also appropriate. A few specific comments are listed below:

1) “The differential binding of NGR to CD13 isoforms” is very important for NGR-based cancer imaging and/or therapy, and thus the correlated discussion (line 484 ~ 490) is strongly suggested to be moved to the Section 2 (2. Aminopeptidase N (APN/CD13) and asparagine–glycine–arginine (NGR) motif);

2) “virtue of the higher specificity” (Line 95) may not be accurate, since heterodimeric radiotracers may detect more normal organs. For example: a normal organ expressing receptor-A can be virtualized by a heterodimer targeting both receptor-A and -B, but not by the monomeric counterpart that targets receptor B only;

3) We understand that due to the avidity effects, heterodimer can prolong tumor-retention; but why dual-targeting agents would have “prolonged circulation time” (Line 99)?

4) Line 250 & 260, the “KHC mice” might should be “KCH mice” ?

Reviewer 2 Report

Trencsényi et al. discuss the importance and feasibility of NGR as a targeting motif for neo-angiogenesis. Looking only into preclinical studies published in the recent years, the authors review the achievements of NGR heterodimers and their potential as a diagnostic and therapeutic tool. Indeed, NGR and other agents that target specific integrin domains have gain popularity in the past years, as seen by the recent increase in published reviews on this topic. Despite the interesting topic selected by the authors, the collective gain generated by this manuscript is somewhat limited when put next to the multitude of other published reviews on the same topic.

To the present manuscript, I have several major concerns:

1.      The introduction does not introduce the reader into nuclear imaging techniques; however, it is discussed that a transition from Ga would be beneficial. For a general reader of the journal, this would not be straight forward. Why is Ga then used? What are the downsides of Ga?

2.      As the authors had to restrict to only the most recent publications for this review, and therefore, despite having more than 100 references, Point 3.1 (i.e., 4 pages) are spent on a detailed presentation of the data and discussing only 3 research articles. I understand the reason behind this choice, however as a reader, if I am interested in so much detail, I would rather read the paper themselves. A similar comment would be again for Point 3.3. Page 12 is mostly spent on a detailed description of ref 66. In my opinion this should be more condensed; for a review, one should not be interested in the %ID/g of a tracer, if it is not in a direct comparison to a different one, in order to draw the conclusion that one is better than the other.

3.      After a detailed review with more the 100 ref and 14 pages, the conclusion is incredibly general. Replace NGR-heterodimers, with any other tracer and the statement would still be true. After so much investment I would prefer to see a more specific conclusion for this manuscript. Is there one specific probe that seems to work better? Is there something we can learn after more the 10 years of developing integrin tracers? Does one specific dimer combination make more sense for an improved development of diagnostic and therapeutic probes?

Reviewer 3 Report

The manuscript of Trencsényi and co-workers entitled:” Radiolabelled NGR-based Heterodimers for Angiogenesis Im-2 aging: A Review of Preclinical Studies” reports the development of new asparagine-glycine-arginine (NGR) heterodimeric probes in preclinical studies to show their efficacy as potential diagnostics and therapeutics platforms.

The review is divided into four sections: the introduction helps the reader to have the main background related to the described topic and the main information about NGR-based probes and how they work. Moreover, the superiority of peptide-based diagnostic compared to full antibody is well articulated and explained, the same is for the multimeric peptide versus their monomers.

Section 2 focuses on different works about NGR motif and their uses in the pre-clinical area highlighting the main radionuclides utilised to label the different radiopharmaceuticals. Figure 1 is clear and helps the reader to understand the strategies utilised to modify the main NGR motif.

Section 3 is divided in other 4 sub-section, the aim is to classify the use of the several NGR probes such as angiogenesis imaging, the ones containing hyaluronic acid, the ones for fusion proteins and compared the different research works. While the table are really useful to underline the main features of different research works, in the text some other figures where the different data are compared are missing. As example at line 169-170 where Trencsényi and co-workers compare the different uptake of [68Ga]Ga-NGR-RGD probe in cell lines the presence of a figure reporting the most important data would be beneficial for the reader. The number of figures and which point the authors want to highlight are of their discretion.

At line 476 please convert the microcurie in megabecquerel since curie are no more accepted from the radiochemist community as main unit of radioactivity.

I believe that the authors did a good work that would be more well-explained after the introduction of the figures.

Reviewer 4 Report

The review "Radiolabeled NGR-based Heterodimers for Angiogenesis Imaging: A Review of Preclinical Studies" describes previous approaches to aminopeptidase N (APN)-specific targeting of tumors using heterobivalent ligands that bind APN as a target as well as to other target structures.

In principle, heterobivalent ligands capable of binding to multiple targets have been shown over the last 10 years to be able to address tumors with significantly higher selectivity, specificity, and affinity than their monovalent monospecific counterparts, since they can bind not only to one target structure present on the tumor, but to several. This aspect is also taken up and discussed in the review and I fully agree with it. However, there are some aspects that are not favorable for a review:

-       - there is not a single molecular formula in the entire review, so one would have to read all the primary literature to understand what exactly was done and which compounds are actually discussed; the properties of such compounds, however, depend largely on their molecular design

-      - that one would have to read up on the primary literature would be of limited concern, however, since it is only about two handfuls of papers that are discussed at all. Therefore, I have the very basic question whether such a topic, for which there is so little literature available and which then also includes so many different molecular designs (conjugates with RGD peptide, with a small molecule or fusion proteins), deserves a review paper at all; it may well be that this topic will become more relevant in the next few years, but that too only when it is understood how exactly such compounds have to be constructed in order to achieve the objective of highly specific and sensitive imaging of malignant diseases. However, since there is little data available so far, this is more a small collection of unrelated examples from which one cannot infer a structure-activity relationship or other relevant correlations

-       - the tables that have been included hardly convey any content that contributes to a higher comprehensibility or to more knowledge, but is rather a summary of experimental details that do not help to discuss relevant things

-       - although I have been dealing with the topic myself for several years, after reading this paper I do not understand how and why and by which mechanism the 3-amino acid peptide binds to APN and how strongly it is expressed in the tumors, i.e. how exactly is the pathway by which the accumulation is supposed to be achieved

-          - although the results of the studies seem to be for the most part rather modest and not very encouraging (the data are not presented / discussed in full, but only single aspects are singled out, so again the primary literature would have to be read to get an impression of the full pharmacokinetics of the compounds), this is not discussed appropriately, as I would expect in a review, but it is only highlighted that the approach investigated in each case is better than the monospecific references (as far as applicable); this is not an adequate and critical discussion of the topic. Furthermore, an adequate discussion includes completeness of presentation of the data, especially when it comes to so few examples being discussed at all. In addition, positive and less positive aspects should then be decisively worked out and compared with other radiopharmaceuticals.

   Overall, I personally find the review therefore not very useful, because it does not inform the reader extensively about the topic, which, moreover, is not well chosen.

No concerns here, only minor editing is required

Round 2

Reviewer 2 Report

The manuscript is significantly improved when compared to its original.